# The Relationship between Health Expenditures and Economic Growth in EU Countries: Empirical Evidence Using Panel Fourier Toda–Yamamoto Causality Test and Regression Models

**DOI:** 10.3390/ijerph192215091

**Published:** 2022-11-16

**Authors:** Ayfer Ozyilmaz, Yuksel Bayraktar, Esme Isik, Metin Toprak, Mehmet Bilal Er, Furkan Besel, Serdar Aydin, Mehmet Firat Olgun, Sandra Collins

**Affiliations:** 1Department of Foreign Trade, University of Kocaeli, İzmit 41650, Turkey; 2Department of Economics, Istanbul University, Istanbul 34452, Turkey; 3Department of Opticianry, Malatya Turgut Ozal University, Malatya 44700, Turkey; 4Department of Economics, Istanbul Sabahattin Zaim University, Istanbul 34303, Turkey; 5Department of Computer Engineering, Harran University, Sanliurfa 63050, Turkey; 6Department of Public Finance, Sakarya University, Sakarya 54187, Turkey; 7School of Health Sciences, Southern Illinois University Carbondale, Carbondale, IL 62901, USA; 8The Department of Technology Transfer, Kastamonu University, Kastamonu 37150, Turkey

**Keywords:** health expenditures, economic growth, regression models, causality

## Abstract

The aim of this study is to investigate the effect of health expenditures on economic growth in the period 2000–2019 in 27 European Union (EU) countries. First, the causality relationship between the variables was analyzed using the panel Fourier Toda–Yamamoto Causality test. The findings demonstrate a bidirectional causality relationship between health expenditures and economic growth on a panel basis. Secondly, the effects of health expenditures on economic growth were examined using the Random Forest Method for the panel and then for each country. According to the Random Forest Method, health expenditures positively affected economic growth, but on the country basis, the effect was different. Then, government health expenditures, private health expenditures, and out-of-pocket expenditures were used, and these three variables were ranked in order of importance in terms of their effects on growth using the Random Forest Method. Accordingly, government health expenditures were the most important variable for economic growth. Finally, Support Vector Regression, Gaussian Process Regression, and Decision Tree Regression models were designed for the simulation of the data used in this study, and the performances of the designed models were analyzed.

## 1. Introduction

Health and economic growth often interact with each other. As the development level of a society rises, it is expected that the health indicators will improve. Furthermore, as health expenditures of the countries increase, infant and child mortality can decrease, and average life expectancy can increase. This can significantly increase labor productivity [1]. Increasing labor productivity is a direct determinant of economic growth.

The health sector has been seen as a sector that supports human capital investments due to its effects on public health in recent years. A common view today is that health can affect economic growth through channels, such as productivity and innovation [2]. This is because the socioeconomic development level of a country is considerably dependent on the health status of individuals [3].

A strengthened health system minimizes production losses caused by diseases. In addition, effective health policies increase the productivity and welfare of the poor, who often may not be able to afford health services, by improving their health conditions [4]. From this point of view, it can be said that health expenditures are not only a consumption good but also an investment good [5].

There is no consensus on the theoretical approaches discussing the effect of health expenditures on growth. The general view is that health expenditures positively affect growth through human and physical capital, which will positively affect the health conditions of individuals. However, according to some studies, health expenditures can cause a cumulative increase in the elderly population by improving the living conditions of the more elderly population. This can be a burden on the economy in the long run. In addition, health expenditures can crowd out more productive public investments. Therefore, the impact of health expenditures on economic growth is not clear [6,7].

The income levels of countries are one of the determining parameters in the effect of health expenditures on growth because high-income countries have relatively better health conditions. For this reason, expenditures on the health system can be seen as consumption expenditures that have no effect on economic growth in these countries. Therefore, under these conditions, health expenditures can negatively affect economic growth [8].

In addition to the role of the health system in growth, economic growth can also be a determinant of health. The common view is that economic growth improves public health because economic growth is one of the prominent indicators of living conditions. In addition, economic growth increases technological progress, including medical technology development and technological advances in health, which also positively affect health outcomes [9,10,11]. Therefore, there may be a unidirectional relationship between health expenditures and growth, or there may be a feedback relationship between the variables [12].

In this study, the relationship between health expenditures and economic growth was investigated in 27 EU countries during 2000–2019. The relationship between health expenditures and growth was investigated from many perspectives for the EU countries. Firstly, the relationship between the two variables was investigated both with the current causality test, and the coefficient estimation was used for the direction of the relationship, and the results were given both for the panel and for the country. Secondly, health expenditures were categorized as public health expenditures, private health expenditures, and out-of-pocket health expenditures, and these expenditures were ranked according to their importance on growth. As far as we know, there is no study in this sample that discusses the relationship between health expenditures and growth simultaneously with all these empirical methods. The study is expected to contribute to the literature in this sense.

The remainder of this research article is structured by sections. Section 2 provides a review of the literature about health and economic growth; Section 3 specifies the data and methodology, and Section 4 provides an overview pertaining to the results of study.

## 2. Literature Review

There is extensive literature on the relationship between health expenditures and economic growth. Some studies use only general government expenditures, and some use different health expenditures, such as out-of-pocket, government, and private expenditures. The findings in the studies differ according to the model, variable, year, or countries used, and different channels of transmission are emphasized in obtaining the findings. According to empirical studies, generally, health expenditures can affect economic growth through two channels. The first is that increased health expenditures reduce productive public investment leading to the crowding-out effect. Therefore, this channel causes health expenditures to affect economic growth negatively. The second channel is that health expenditures positively affect the productivity of the workforce by improving the health status of employees. According to this channel, health expenditures increase economic growth. Therefore, the impact of health expenditures on economic growth will likely depend on these two channels and their relative strength [7].

There are four approaches in studies investigating the relationship between health expenditures and economic growth using causality analyses. These are income, health, feedback hypothesis, and neutrality hypothesis. The first is the income approach, which suggests that there is a unidirectional causality from economic growth to health expenditures. The second is the health approach, which emphasizes the existence of unidirectional causality from health expenditures to growth. The third is the feedback hypothesis, which suggests that there is a bidirectional causal relationship between health expenditures and growth. The fourth is the neutrality hypothesis, which suggests that there is no causality between the two variables [13,14,15].

Alhowaish [16] and Chang et al. [17] reached findings that support the income approach, which suggests that there is unidirectional causality between economic growth and health expenditures in Saudi Arabia and China. Yumuşak and Yildirim [18] and Mehrara and Musai [19] obtained results that support the health approach, which emphasizes the unidirectional causality between health expenditures and economic growth in Turkey and Iran. Çetin and Ecevit [20] reached findings supporting the neutrality hypothesis. According to the study, there is no statistically significant relationship between public health expenditures and growth in 15 OECD countries. According to Fendoğlu and Gökçe [21], who reached similar findings, there is no long-run relationship between economic growth and health expenditures in Turkey.

There are many studies supporting the feedback hypothesis suggesting that there is a bidirectional relationship between health expenditures and growth. For example, Wang et al. [22], Şen and Bingöl [23], Nasreen [24], Saraçoğlu and Songur [25], Chaabouni and Zghidi [13], Nasiru and Usman [26], Sethi et al. [27], and Aydemir and Baylan [28] found a bidirectional causality relationship between health expenditures and economic growth in Pakistan, Turkey, 20 Asian economies, 10 Eurasian countries, 51 countries, Nigeria, South Asian countries, and Turkey.

In some studies, the relationship between health expenditures is investigated with both causality and different econometric models. For example, Çelik [29] suggested that a unidirectional causality relationship exists between economic growth and health expenditures in G20 countries. In addition, his research indicates that health expenditures positively affect economic growth. Üzümcü and Söğüt [30] found a cointegration relationship between public health expenditures and growth in Turkey. According to the study, there is a bidirectional Granger causality relationship between the two variables, and public health expenditures positively affect growth. Ifa and Guetat [2] found a cointegration relationship between economic growth and public health expenditures in Tunisia and Morocco. In addition, public health expenditures affect growth positively in both countries, and there is a positive bidirectional causality relationship between the variables in both countries. Demirgil et al. [31] found a long-term cointegration relationship between health expenditures and economic growth, and health expenditures affect economic growth positively in Turkey. In the study, which also included a causality analysis, a unidirectional causality relationship was found between economic growth and health expenditures. Uslu [32] found a cointegration relationship between health expenditures and economic growth in 36 OECD countries. On the other hand, health expenditures affect growth positively, and there is bidirectional causality between health expenditures and growth in these countries. Akinci and Tuncer [33] found a long-term relationship between health expenditures and economic growth in Turkey. Furthermore, a bidirectional causality between economic growth and health expenditure was found.

There are many studies that suggest a positive relationship between health expenditures and economic growth exists. For example, Aboubacar and Xu [34] found a positive and statistically significant relationship between health expenditures and economic growth in Sub-Saharan Africa. This is because healthcare is a necessity, not a luxury. Hayaloğlu and Bal [35] argued that total health expenditures, public health expenditures, and private health expenditures positively affect economic growth in 54 upper middle-income countries. Hatam et al. [5] suggested that there is a positive and strong relationship between health expenditures and economic growth in ECO countries. According to the study, the findings indicate that health expenditures are an investment good rather than a consumer good. Furthermore, Selim et al. [36] found a positive relationship between health expenditures and economic growth in both the short run and long run in 27 EU countries and Turkey. Gaies [37] found that public and private domestic health expenditures positively affect income in 60 developing countries, and this effect becomes stronger as the level of human and physical capital increases. Raghupathi and Raghupathi [38] investigated the relationship among public health expenditures, labor productivity, income, and GDP in the US. According to the study, public health expenditures are positively related to personal expenditures, labor productivity, and GDP. Therefore, the increase in public health expenditures is positively related to economic performance. On the other hand, Kizil and Ceylan [39], Sheikh et al. [40], and Zhang et al. [41] found that public health expenditure increases economic growth in Turkiye, Pakistan, and China. In addition, Somé et al. [42], Sahnoun [6], Yildiz and Yildiz [43], Başar et al. [44], Safdari et al. [45], Modibbo and Saidu [46], and Penghui et al. [47] found that health expenditures positively affect economic growth in 48 African countries, Tunisia, 47 European and Central Asian countries, Turkey, Iran, 45 African countries, and China.

In some studies, the findings differ depending on parameters, such as period, method, or country. For example, Konat [48] found a unidirectional causality relationship between economic growth and health expenditures in 17 OECD countries. However, there is a bidirectional relationship between the variables in negative shocks. Kizilkaya [49] suggested that there is a unidirectional causality relationship between economic growth and health expenditures in 21 OECD countries. On the country basis, there is a unidirectional causality relationship between economic growth and health expenditures in Austria, Denmark, Finland, Germany, Korea, Spain, Sweden, Switzerland, and the Netherlands, but the causality relationship is bidirectional for Iceland. Odhiambo [50] divided Sub-Saharan African countries into two groups according to low-income and middle-income countries. He also categorized health expenditures as private and public expenditures. According to the study, when public health expenditures are used, there is unidirectional causality between health expenditures and economic growth in low-income countries, but there is no causality between variables in middle-income countries. However, when using private health expenditures, there is short-run causality between economic growth and health expenditures in middle-income countries; however, there is no causal relationship between variables in low-income countries. Keyifli and Recepoğlu [51] suggested that there is no causal relationship between growth and health expenditures in E7 countries, but there is a causality relationship between economic growth and health expenditures in India and Indonesia. Mojahid et al. [52] analyzed the relationship between public health expenditures and growth in Bangladesh, India, and Nepal. According to the study, there is unidirectional causality between growth and public health expenditures in Bangladesh and Nepal in the short run, but there is bidirectional causality between the variables in India. In the long-run, there is unidirectional causality between public health expenditures and growth in all three countries. Penghui [47] found that financial inputs and health insurance expenditures positively affect economic growth within provinces and neighboring provinces in China. Health personnel input may positively affect the growth of provinces, but it is not effective in the growth of neighboring provinces.

Tiraş and Ağir [1] analyzed the causal relationship between income and health expenditure components in OECD countries. Accordingly, a causal relationship is observed between at least one of the health expenditure components and income in 28 OECD countries, but there is no causal relationship between income and any of the health expenditure components in eight OECD countries. Akar [53] found that there is a significant relationship between health expenditures, the relative price of these expenditures, and economic growth in Turkey in the long run, but there is no significant relationship between the variables in the short run. Balaji [12] concluded that there is no long-run relationship between health expenditures and economic growth in the four Southern States of India (Andhra Pradesh, Karnataka, Kerala, and Tamil Nadu). However, there is unidirectional causality between economic growth and health expenditures in Andhra Pradesh. According to Rivera [4], public health expenditures positively affect economic growth in Spanish regions, but public investments in health do not affect productivity because the impact of both health and education infrastructures takes much longer to affect productivity than other types of infrastructure.

Different from these studies, Eggoh et al. [54] found that public health expenditures negatively affect economic growth in 49 countries. In addition, Bats [8] found that the effect of health expenditures on growth is negative, but this effect is slightly more in 17 relatively wealthy OECD countries. According to the study, there could be two reasons for this. First, health expenditures must have an impact on the health of a population in order to affect the economy. However, since this will take time, it is difficult for health expenditures to affect economic growth in the short term. Second, the countries in the sample are relatively wealthy OECD countries. In general, the wealthier a country is, the healthier its population. Thus, the marginal benefit of health expenditures in these countries is relatively small compared to poorer countries. Thus, using a sample of relatively healthy populations reasonably explains why health expenditures have a coefficient value close to zero. However, when a country’s population is already relatively healthy, health expenditures can often be viewed as consumption that increases the population’s utility but has no effect on growth. In this case, because the money spent on health expenditures can be considered an ineffective investment, the impact of health expenditures on growth may be negative.

The available literature that investigates the relationship between health expenditures and growth with a threshold value is quite limited. One of these studies is Wang’s [7] article on OECD countries. According to the study, when the ratio of health expenditures to GDP is less than the optimal level of 7.55%, increases in health expenditures lead to a higher economic performance. Therefore, for economic development, governments should increase health investments until they reach the optimal level. Yang [55] determined the level of human capital as the threshold and investigated the relationship between public health expenditures and economic growth for different levels of human capital in 21 developing countries. According to the study, when the level of human capital is low, public health expenditures negatively affect economic growth. When human capital is at a medium level, public health expenditures positively affect economic growth, but this effect is slight. However, when the level of human capital is high, the positive effect of public health expenditures on economic growth increases significantly.

## 3. Materials and Methods

### 3.1. Data

In this study, the relationship between health expenditures and economic growth was investigated in 27 EU countries during the 2000–2019 period. Annual data were used in the study. Details of the variables used are given in Table 1.

The time dynamics of the variables are given in Figure 1.

Descriptive statistics are presented in Table 2.

### 3.2. Methods

In this study, the relationship between health expenditures and economic growth was investigated using causality analysis and machine learning methods. The relationship between the variables was examined with the Panel Fourier Toda–Yamamoto causality test. For other estimations, Support Vector Algorithm (SVR), Gaussian Process Regression (GPR), Decision Tree Regression, and Random Forest Model methods from machine learning methods were used.

#### 3.2.1. Panel Fourier Toda–Yamamoto Causality Test

In the study, the causality relationship between health expenditures and economic growth was investigated with the causality test developed by Yilanci and Gorus [56] and named as Panel Fourier Toda–Yamamoto (PFTY). The null hypothesis for this test was that there was no causality. To obtain the results, the bivariate panel VAR Model shown below was estimated [56].
(1)$$y_{i,t}=\mu_{i}+\overset{k_{i}+d_{max_{i}}}{\underset{j=1}{\sum }}A_{11}y_{i,t-j}+\overset{k_{i}+d_{max_{i}}}{\underset{j=1}{\sum }}A_{12}x_{i,t-j}+A_{13}\sin\left(\frac{2\pi tf_{i}}{T}\right)+A_{14}\cos\left(\frac{2\pi tf_{i}}{T}\right)+u_{i,t}$$
(2)$$x_{i,t}=\mu_{i}+\overset{k_{i}+d_{max_{i}}}{\underset{j=1}{\sum }}A_{21}y_{i,t-j}+\overset{k_{i}+d_{max_{i}}}{\underset{j=1}{\sum }}A_{22}x_{i,t-j}+A_{23}\sin\left(\frac{2\pi tf_{i}}{T}\right)+A_{24}\cos\left(\frac{2\pi tf_{i}}{T}\right)+u_{i,t}$$

In the equations, *t* = 1, 2, 3, ……, *T*, and *i* = 1, 2, 3, ……, *N*. *k* represents the optimum lag order with the help of information criteria, and dmax represents the maximum degree of integration. The terms with sin and cos show the Fourier function. Equations (1) and (2) were estimated to test the validity of the null hypothesis stating that there is no causality. The first equation shows that there is no causality from the dependent variable to the independent variable, and the second equation shows that there is no causality from the independent variable to the dependent variable. In the study, income per capita (lnGDP) was used as the dependent variable to represent growth, while the share of health expenditures in GDP (lnHE) as an independent variable was used as an indicator of health expenditures. The estimation results can be obtained on the basis of the country as well as the panel. The bootstrap method was used to obtain test statistics. The Fisher’s test statistic was calculated with the assistance of the equation below [56].
(3)$$FTYP=-2\overset{N}{\underset{i=1}{\sum }}ln\left(p_{i}^{*}\right)$$
$p_{i}^{*}$, the bootstrap corresponding to the Wald statistic, shows the *p*-value. This value was calculated separately for each cross-section. The degree of stationarity of the series was not important in using the test. This test allowed series to be I(0) or I(1), but took into account Cross-Section Dependency. This test considered structural changes [56].

In the study, the correlation between units was examined, and it was determined which Unit Root test would be used according to the results obtained. After determining the stationarity levels of the series, the causal relationship between health expenditures and growth was examined.

The CD test developed by Pesaran [57] and the Bias_Adjusted LM test developed by Pesaran, Ullah, and Yamagata [58] were used to examine the Cross-Section Dependency. The CIPS test developed by Pesaran [59] and the Z_A_^SPC^ and Z_A_^LA^ Unit Root tests developed by Hadri and Kruzomi [60] were used to test the stationarity of the series.

#### 3.2.2. Support Vector Regression

The Support Vector Algorithm (SVR) was used in regression, although it was originally an algorithm for classification. Some data problems were solved using these two models. An adaptation of Support Vector Machines, which are extensively used for classification problems, for regression was proposed by Smola et al. [61]. The basic logic of SVR is to try to ascertain the regression function that will lessen the expected risk error instead of reducing the training error. In SVR, the goal is to determine the line or curve so that the maximum point in the range of a margin can be taken with the smallest error. When Support Vector regression is applied, it is to make certain that the range drawn will include the maximum point. The points where these drawn maximum intervals intersect are called support points. In SVR, kernel functions allow it to search a wide range of solutions. Typical examples of kernel function are a form of linear, polynomial, and Gaussian. Below, the formulas of Gaussian, polynomial and linear kernels are given in Equations (4)–(6) [62].
(4)$$\mathrm{Gaussian}\;\mathrm{kernel}:\;(\;,\;x_{j})=exp\;\left(\frac{\|x_{i}-x_{j}{\|}^{2}}{2\sigma^{2}}\right)\;$$
Polynomial kernel: *Q*(*x*_*i*_, *x*_*j*_) = (1 + *x*_*i*_ · *x*_*j*_)^*d*^(5)
(6)$$\mathrm{Linear}\;\mathrm{kernel}:\;Q(x_{i},\;x_{j})=x_{i}^{t}x_{j}\;$$

In this study, experiments were carried out on three different SVR kernel models, namely Gaussian, polynomial, and linear. However, the Gaussian kernel was used because it was obtained from the most successful Gaussian kernel.

#### 3.2.3. Gaussian Process Regression

Gaussian Process Regression (GPR) is a kernel-based, nonparametric probabilistic model, and the basic principles were determined by Rasmussen [63]. GPR is a model that uses the Bayesian regression approach and is suitable for solving nonlinear regression problems [64]. The GPR Model is a method that works well on small datasets and is capable of measuring uncertainty of predictions. It is suitable for solving nonlinear regression problems. GPR has four different core models. These are Rational Quadratic, Square Exponential, Matern 5/2, and Exponential Model. Formulas of Rational Quadratic, Square Exponential, Matern 5/2, and Exponential Models are given below in (7)–(10).
(7)$$k_{RQ}\left(x,x^{\prime }\right)=\sigma^{2}{\left(1+\frac{{\left(x-x^{\prime }\right)}^{2}}{2\alpha l^{2}}\right)}^{-\alpha }$$
(8)$$k_{SE}\left(x,x^{\prime }\right)=\sigma^{2}\exp\left(\frac{-{\left(x-x^{\prime }\right)}^{2}}{2l^{2}}\right)$$
(9)$$k_{M_{v}}\left(x,x^{\prime }\right)=\sigma^{2}\frac{1}{2^{v-1}\tau \left(v\right)}{\left(\sqrt{2V}\frac{\left|x-x^{\prime }\right|}{l}\right)}^{v}\cdot B_{v}\left(\sqrt{2V}\frac{\left|x-x^{\prime }\right|}{l}\right)$$
(10)$$k_{E}\left(x,x^{\prime }\right)=\sigma^{2}\exp\left(\frac{\left|x-x^{\prime }\right|}{l}\right)$$

In this study, experiments were also carried out on four SVR kernel models, namely Rational Quadratic, Square Exponential, Matern 5/2, and Exponential models. The most successful result was obtained from the Rational Quadratic Model.

#### 3.2.4. Decision Tree Regression

Decision Trees are in the form of a tree structure that can be built on both Regression and Classification models. If the target features are approximated based on the Classification or Regression processes with the Decision Tree method consisting of discrete data or certain categories, the model used is called the Classification Tree. If the feature data are comprised of continuous variables, they are identified as the Model Regression Tree [65]. In a simple Regression Tree structure, there are three basic elements defined as node, branch, and leaf. In the structure in question, a node represents each attribute. The top part of the tree structure includes roots, and the lowest part consists of leaves. The parts that remain between the root and the leaves and provide the relationship between the upper nodes and the lower nodes are expressed as branches [66].

#### 3.2.5. Model Evaluation Metrics

The performance of the model was tested using Mean Square Error (MSE), Root Mean Square Error (RMSE), Mean Absolute Error (MAE), and *R*^2^ Score. These metrics are the main performance metrics used in performance evaluations of regression models. MSE gives an absolute result on how much the predicted results differ from the actual number. RMSE is the square root of MSE. It is used more often than MSE because sometimes the MSE value can be too large to be easily compared. Therefore, the MSE was calculated with the square of the error, thus facilitating interpretation. The MAE sums the absolute error value and returns the middle of the absolute difference between the model prediction and the target value. R^2^ is a statistical measure of how close the data are to the fitted regression line. The equations of these performance measures are given below in (11)–(14).
(11)$$\mathrm{MSE}\left(y,\hat{y}\right)=\frac{1}{n_{samples}}\overset{n_{samples}-1}{\underset{i=0}{\sum }}{\left(y_{i}-{\hat{y}}_{i}\right)}^{2}$$
(12)$$\mathrm{RMSE}\left(y,\hat{y}\right)=\sqrt{\mathrm{MSE}\left(y,\hat{y}\right)}$$
(13)$$\mathrm{MAE}\left(y,\hat{y}\right)=\frac{1}{n_{samples}}\overset{n_{samples}-1}{\underset{i=0}{\sum }}\left|y_{i}-{\hat{y}}_{i}\right|$$
(14)$$R^{2}\left(y,\hat{y}\right)=1-\frac{\sum_{i=0}^{n1}{\left(y_{i}-{\hat{y}}_{i}\right)}^{2}}{\sum_{i=0}^{n1}{\left(y_{i}-{\hat{y}}_{i}\right)}^{2}}$$

#### 3.2.6. Random Forest Model

The Random Forest Model is useful in solving two kinds of problems: (i) establishing a prediction rule for a supervised learning issue and (ii) assessing and ranking variables relative to their potential to predict a solution. Random Forest, like neural networks and other nonlinear classifiers, is nonlinear. It is commonly used to sort nonlinearly separable data. The latter is achieved using the Random Forest algorithm’s Variable Importance measures, which are produced for each predictor automatically. The Random Forest Variable Importance Model is recognized as being able to successfully identify predictors involved in interactions, that is, predictors that can only predict the response in conjunction with one or more other predictors [67,68,69]. A Random Forest is a set of tree predictors where *x* represents the observed input (covariate) vector of length *p,* and the *θ_k_* are independent and identically distributed *(iid)* random vectors, as given below.
*h(x, θ_k_)*, *k* = 1, … *K*(15)

As previously stated, this study concentrated on regression problems where the outcome was numerical. With *y*, we did come across certain classification (categorical outcome) issues. The (training) data observed were thought to be independently drawn from the (*X*, *Y*) joint distribution and consisted of *n*(*p* + 1)-tuples (*x*_1_, *y*_1_), ….. (*x_n_*, *y_n_*).

The unweighted average over the collection was the Random Forest prediction for regression: (16)$$h\left(x\right)=\left(1/K\right)\sum_{k=1}^{K}h\left(x;\theta k\right)$$

As k→∞ the Large Numbers’ Law provides,
(17)$$E_{X,Y}{\left(Y-\bar{h}\left(X\right)\right)}^{2}\to E_{X,Y}{\left(Y-E_{\theta }h\left(X;\theta \right)\right)}^{2}$$

The prediction (or generalization) error for the Random Forest, labeled as $\;PE_{t}^{*},\;$ is seen as the value on the right. Random Forests do not overfit due to the convergence in (14). The average single tree prediction error noted as *h*(*X*; θ) is defined below:(18)$$PE_{t}^{*}=E_{\theta }E_{X,Y}{\left(Y-h\left(X;\theta \right)\right)}^{2}$$

We assumed that the tree is unbiased for all, i.e., $EY=E_{X}h\left(X;\theta \right).$
(19)$$PE_{f}^{*}\leq \bar{\rho }PE_{t}^{*}$$

Then, $\bar{\rho }\;$ is the weighted correlation between the residuals *Y*-*h*(*X*; θ) and *Y*-*h*(*X*;$\;\theta^{\prime }$) for the independent variables $\theta ,\theta^{\prime }$.

The inequality (19) identifies what is needed for precise Random Forest regression: (i) there is a low correlation between the residuals of different forest tree members and (ii) the individual trees have a low prediction error. Furthermore, it is expected that the Random Forest will reduce the individual tree error, $PE_{t}^{*}$, by the factor. As a result, the injected randomization aims for a low correlation [69,70,71].

The accuracy significance measure is the most sophisticated metric available of variable importance in Random Forests. Its justification is as follows: The predictor variable *X_j_* is severed from its initial link with the response *Y* by randomly permuting it. If the original variable *X_j_* was connected with the response, the prediction accuracy (i.e., the number of observations categorized correctly) reduces significantly when the permuted variable *X_j_* is used to predict the answer along with the remaining unpermuted predictor variables. As a result, the difference in prediction accuracy between before and after permuting *X_j_* is a decent indicator of a variable’s importance. When compared to other variables, a variable with a higher significance score is one that is more crucial for categorization [72].

## 4. Results

The obtained empirical findings are given in this section. The results obtained from the Cross-Section Dependence and Unit Root tests are given in Table 3.

Cross-Section Dependence test results indicate that there is Cross-Section Dependence in all series. On the other hand, different results were obtained compared to the unit tests. But as a result, all series were either I(0) or I(1), so the panel Fourier causality test was performed, and the results of the causality test are given in Table 4.

According to the panel Fourier causality test results, there was a bidirectional causality relationship between health expenditures and economic growth on the basis of the panel. On the basis of country, there was a bidirectional causality relationship between health expenditures and economic growth in Denmark, France, and Italy, but there was a unidirectional causality relationship between health expenditures and growth in Cyprus, Czech Republic, Estonia, Greece, Hungary, Malta, Slovak Republic, and Slovenia. A unidirectional causality relationship was found between growth and health expenditures in Austria, the Netherlands, Poland, Portugal, and Sweden. In other countries, there was no causal relationship among the variables.

To explain the impact of health spending on economic growth in each member of the EU, the Random Forest Model was used. The estimation performances of the developed models were calculated using RMSE, and the R2 scores and the results were almost 0 and 1, respectively, and the analysis findings are presented in Table 5.

According to the general panel findings, health expenditures positively affected economic growth. On the country basis, although the findings were different, this effect was mostly positive. Health expenditures negatively affected economic growth in Austria, Croatia, Greece, Italy, Portugal, and Spain, but positively affected growth in 21 other countries.

When looking at other control variables, in general, we saw that trade was the strongest variable for economic growth, and urban population was the second most important variable for economic growth, and these variables affected economic growth positively in all countries. Inflation affected economic growth positively in some countries and negatively in others, but this effect was slight. Employment affected economic growth positively in all countries except Cyprus.

In addition, health expenditures were categorized as public health expenditures, private health expenditures, and out-of-pocket health expenditures, and these expenditures were ranked according to their importance for growth, and the test results are presented in Table 6.

According to Table 6, the R^2^ and RMSE values, which determine the performance of the model, were obtained as 1 and 1.151 × 10^−13^, respectively. As seen in Table 6, public health expenditures were the strongest variable for economic growth. On the other hand, private health expenditures were the weakest variable for economic growth.

Models in the study were developed using the relevant tools and libraries of MATLAB software, MathWorks, Portola Valley, CA, USA. The dataset containing 540 samples was reserved for testing and training using 10-fold cross validation. Before the models were built, the process was applied to all the data in the data set. SVR, GPR, and DTR were used as models [73]. The estimation performances of all developed models were calculated using the MSE, RMSE, MAE, and R^2^ scores. The accuracy of the models was very sensitive to the selection of the parameters, and there are studies on hyperparameter optimization in the literature, but there is no mathematical model to obtain the desired values of these parameters. In this study, different hyperparameters were tried for the SVR, GPR, and DTR Models, and the parameters that gave the best results were selected. The results for each model are given in Table 7, Table 8 and Table 9.

The results obtained from the SVR Model are given in Table 10. The highest MSE value was 0.93881 in the first fold; the highest RMSE value was 0.93684 in the fifth fold; the highest MAE value was 0.92991 in the tenth fold, and the highest R^2^ score was 0.93683 in the ninth fold. The mean MSE, RMSE, MAE, and R^2^ score values obtained from the SVR Model were 0.90397, 0.90116, 0.89317, and 0.90637, respectively.

In the results obtained from the GPR Model as shown in Table 11, the highest MSE value was 0.90514 in the fifth fold; the highest RMSE value was 0.90643 in the seventh fold; the highest MAE value was 0.91844 in the first fold, and the highest R^2^ score was 0.90888 in the fourth fold. The mean MSE, RMSE, MAE, and R^2^ score values obtained from the GPR Model were 0.86060, 0.86502, 0.85825, and 0.86892, respectively.

In the results obtained from the DTR Model as shown in Table 12, the highest MSE value was 0.88851 in the first fold; the highest RMSE value was 0.89817 in the eighth fold; the highest MAE value was 0.89939 in the first fold, and the highest R^2^ score was 0.91184 in the ninth fold. The mean MSE, RMSE, MAE, and R^2^ score values obtained from the DTR Model were 0.85496, 0.86046, 0.86932, 0.85372, respectively.

## 5. Conclusions

The relationship between health expenditures and economic growth is one of the important debates. The fact that health expenditures are in a form that will strengthen the health system contributes positively to the strengthening of the institutional infrastructure for health and the improvement of the health conditions of individuals. A healthier society has a decisive role on economic growth through both human capital and physical capital. In particular, the limited access of the poor workforce to health services can cause health to be a burden on the economy through the total workforce channel because there is a complementary relationship between health and human capital. Therefore, the efficient use of health expenditures can positively affect economic growth through labor productivity. However, if health expenditures are used inefficiently or do not improve health outcomes, this can lead to a waste of resources. According to another point of view, increased health expenditures can increase the life expectancy of elderly individuals and lead to a burden on the economy. This is one of the debatable parameters, especially in countries with a high elderly population. Another controversial issue is the exclusion effect of health expenditures. Accordingly, an increase in health expenditures can crowd out more productive investments.

In this study, firstly the relationship between health expenditures and economic growth was investigated in 27 EU countries during 2000–2019. The causality relationship between health expenditures and growth was examined using the Panel Fourier Toda–Yamamoto Causality Test. According to the Panel Fourier Toda–Yamamoto Causality Test results, on a panel basis, there was a bidirectional causality relationship between health expenditures and economic growth. On the country basis, there was a bidirectional causality relationship between health expenditures and economic growth in Denmark, France, and Italy. In Cyprus, Czech Republic, Estonia, Greece, Hungary, Malta, Slovak Republic, and Slovenia, and a unidirectional causality relationship was found between health expenditures and economic growth, and there was unidirectional causality between growth and health expenditures in Austria, the Netherlands, Poland, Portugal, and Sweden. In Belgium, Croatia, Finland, Germany, Ireland, Latvia, Lithuania, Luxembourg, Romania, Spain, and Bulgaria, there was no causal relationship between neither health expenditures and economic growth nor economic growth and health expenditures.

Secondly, the effects of health expenditures on economic growth were analyzed using the Random Forest Method. According to the analysis findings, health expenditures positively affected economic growth in EU countries. When the findings were analyzed on a country basis, this effect was generally positive, but in some countries, it was negative. Health expenditures negatively affected economic growth in Austria, Croatia, Greece, Italy, Portugal, and Spain but positively affect growth in 21 other countries. When looking at other control variables, the effect of urban population and trade was more significant according to other variables, and this effect was positive in all countries. Inflation affected economic growth positively in some countries and negatively in others, but this effect was not significant. Employment affected economic growth positively in all countries except Cyprus.

In addition, health expenditures were categorized as government health expenditures, private health expenditures, and out-of-pocket expenditures, and these variables were ranked in order of importance for growth. Accordingly, government health expenditure was the most important variable for economic growth. The fact that public health expenditures mostly covered the general population may have been determinant for obtaining these findings. Thirdly, the performances of the designed models were analyzed. In this way, we could draw future conclusions about the relationship between economic development and health expenditures in various countries prospectively.

The positive effect of health expenditures on growth reveals that the public should also focus on health expenditures, such as education and infrastructure because health expenditures are not only limited to improving the living conditions of individuals, but also feed human and physical capital, which is an important component of production factors, and thus become determinant for economic growth. In addition, considering that the most effective type of health expenditure on growth in the study was public health expenditures, the policies of the public toward the health sector become even more important. In this context, the public’s policies toward health expenditures, which will positively affect social health outcomes, can cause the health sector to be determinant for economic growth as well as social welfare.

## Figures and Tables

**Figure 1 ijerph-19-15091-f001:**
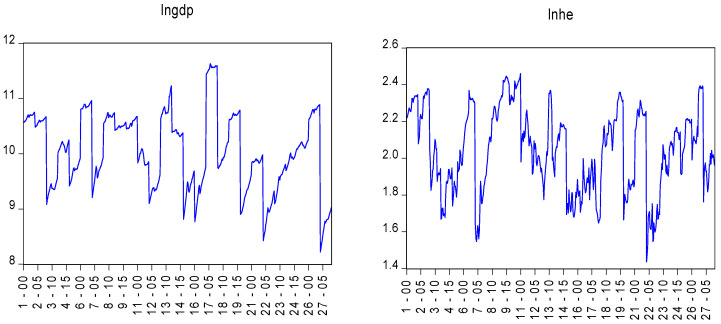
Time Dynamics of Variables.

**Table 1 ijerph-19-15091-t001:** Description of Variables.

Variables	Description	Source
lnGDP	GDP per capita (constant 2015 in USD )	World Bank Open Data
lnHE	Current health expenditures (% of GDP)	World Bank Open Data
INF	Inflation, consumer prices (annual %)	World Bank Open Data
lnEMP	Employment to population ratio, 15+, total (%) (modeled ILO estimate)	World Bank Open Data
lnURB	Urban population (% of total population)	World Bank Open Data
lnTRD	Trade (% of GDP)	World Bank Open Data
lnPOCKET	Out-of-pocket expenditures (% of GDP)	World Bank Open Data
lnPRIVATE	Domestic private health expenditures (% of GDP)	World Bank Open Data
lnGOV	Domestic general government health expenditures (% of GDP)	World Bank Open Data

‘ln’ indicates the natural logarithm of the variables.

**Table 2 ijerph-19-15091-t002:** Descriptive Statistics.

Statistics	lnGDP	lnHE	INF	lnURB	lnEMP	lnTRD	lnGOV	lnPOCKET	lnPRIVATE
Mean	10.039	2.044	2.561	4.259	3.959	4.661	1.700	0.423	0.706
Maximum	11.629	2.459	3.366	4.585	4.148	5.940	2.232	1.277	1.400
Minimum	8.220	1.437	−4.478	3.926	3.630	3.815	0.761	−0.905	−0.821
Std. Dev.	0.696	0.226	45.667	0.175	0.103	0.459	0.297	0.437	0.412
Skewness	−0.084	−0.206	6.364	−0.028	−0.397	0.446	−0.557	−0.351	−0.811
Kurtosis	2.480	2.148	66.493	2.039	2.622	2.778	2.967	2.626	3.410
Obs.	540	540	540	540	540	540	540	540	540

**Table 3 ijerph-19-15091-t003:** Unit Root and Cross-Section Dependency Test Results.

Unit Root Test	Model	lnGDP	∆lnGDP	lnHE	∆lnHE
CIPS	Constant	−2.024	−2.992 ***	−2.269 **	−3.759 ***
	Constant + Trend	−1.918	−3.417 ***	−2.373	−3.874 ***
Z_A_^SPC^	Constant	1.851	1.226 ***	−1.941 ***	−2.458 ***
	Constant + Trend	4.540	0.654 ***	−3.804 ***	−3.341 ***
Z_A_^LA^	Constant	4.314	5.922	−0.764 ***	−2.380 ***
	Constant + Trend	5.445	1.388 ***	−4.298 ***	−3.867 ***
CD Test	Constant	48.469 ***	47.501 ***	24.071 ***	24.157 ***
	Constant + Trend	48.801 ***	48.914 ***	23.932 ***	22.801 ***
LM_Adj_	Constant	144.515 ***	159.577 ***	125.974 ***	104.907 ***
	Constant + Trend	126.484 ***	147.444 ***	112.140 ***	99.574 ***

Maximum lag length was taken as 3. The main hypothesis for the CIPS test was that the series was not stationary, while for Z_A_^SPC^ and Z_A_^LA^, the series was stationary. *** and ** indicate that the series is stationary at 1% and 5% significance levels, respectively, and there is interunit correlation for CD and LM_Adj_.

**Table 4 ijerph-19-15091-t004:** Panel Fourier Causality Test Results.

Country	H_o_: lnHE ↛ lnGDP		H_o_: lnGDP ↛ lnHE	
	Test Statistics	Freq.	Test Statistics	Freq.
Austria	3.2335 (0.4387)	1	16.1680 * (0.0533)	1
Belgium	9.4385 (0.1223)	1	6.8707 (0.2039)	1
Croatia	0.6891 (0.6831)	1	0.6451 (0.6970)	1
Cyprus	12.1535 * (0.0800)	2	3.8802 (0.3724)	2
Czech Republic	7.6274 * (0.0691)	1	0.2331 (0.8979)	1
Denmark	17.5429 ** (0.0453)	2	107.1951 *** (0.001)	2
Estonia	36.0320 *** (0.0095)	1	4.8764 (0.2990)	1
Finland	7.3699 (0.1773)	3	3.8106 (0.3779)	3
France	17.0566 ** (0.0481)	1	13.3507 * (0.0688)	1
Germany	3.8451 (0.2092)	2	0.3977 (0.8206)	2
Greece	23.2997 ** (0.0263)	1	8.5661 (0.1333)	1
Hungary	13.5663 * (0.0700)	1	0.2200 (0.9721)	1
Ireland	0.7612 (0.8520)	3	0.3695 (0.9407)	3
Italy	46.1026 *** (0.0068)	1	16.8175 * (0.0539)	1
Latvia	2.6847 (0.3078)	1	4.9703 (0.1443)	1
Lithuania	0.0042 (0.9462)	2	0.0069 (0.9346)	2
Luxembourg	2.4047 (0.3538)	1	0.8641 (0.6627)	1
Malta	11.7920 * (0.0887)	1	3.6337 (0.3928)	1
Netherlands	5.3293 (0.2672)	1	12.9675 * (0.0751)	1
Poland	1.4357 (0.7077)	1	42.8808 *** (0.0077)	1
Portugal	0.5657 (0.9008)	1	19.3221 ** (0.0331)	1
Romania	3.5326 (0.4147)	1	2.1782 (0.5775)	1
Slovak Republic	11.4697 * (0.0909)	1	3.2798 (0.4380)	1
Slovenia	6.7541 * (0.0853)	1	1.1445 (0.5962)	1
Spain	3.6703 (0.3987)	3	1.8859 (0.6281)	3
Sweden	0.2943 (0.9511)	3	15.5841 * (0.0524)	3
Bulgaria	1.4975 (0.6977)	1	2.4728 (0.5424)	1
Panel Fisher	95.1339 ***		86.1931 **	
Asymptotic *p*-value	0.0004		0.035	
Bootstrap cv (10%)	68.67483		69.7414	
Bootstrap cv (5%)	73.61507		75.0155	
Bootstrap cv (1%)	83.5826		87.4341	

↛ indicates that there is no causality. Value in parentheses indicates *p* values. ***, **, and * indicate that there is a causality relationship at the levels of 1%, 5%, and 10%, respectively.

**Table 5 ijerph-19-15091-t005:** Random Forest Test Results.

	HE	INF	URB	EMP	TRD
Panel	4.20274	0.48353	3.67872	2.52721	3.08333
Austria	−0.04006	−0.06606	0.82953	0.08520	0.76797
Belgium	0.18771	0.05852	0.78308	0.04079	0.84645
Croatia	−0.06063	−0.05951	0.91264	0.19195	0.64494
Cyprus	0.08390	−0.04049	0.42219	−0.04579	0.53095
Czech Republic	0.09470	−0.04747	0.13136	0.19863	1.23124
Denmark	0.14759	0.02179	0.77544	0.00000	0.91575
Estonia	0.16171	−0.04955	0.16357	0.12471	1.12550
Finland	0.05083	−0.03123	0.92039	0.07244	0.67434
France	0.00716	−0.08615	0.48573	0.04124	1.09661
Germany	0.18895	0.01729	0.37307	0.08679	1.44704
Greece	−0.07080	0.10012	0.75836	0.11164	0.54141
Hungary	0.15906	0.05670	1.13889	0.07577	0.56625
Ireland	0.08543	0.00000	0.49588	0.06224	0.96062
Italy	−0.02995	0.06395	1.24267	0.13134	0.22563
Latvia	0.08434	−0.06984	0.09076	0.11756	1.12132
Lithuania	0.07130	−0.02938	0.57670	0.12392	1.10871
Luxembourg	0.01025	0.00000	0.83300	0.02466	0.56320
Malta	0.04945	0.07797	1.58933	0.08273	0.69427
Netherlands	0.09514	−0.07454	0.91621	0.01908	0.62658
Poland	0.12966	−0.08762	1.56791	0.08309	0.49428
Portugal	−0.02523	−0.08611	0.52299	0.08640	0.37435
Romania	0.05065	0.11713	0.88771	0.01094	0.87937
Slovak Republic	0.13799	0.07418	1.54970	0.04477	0.42628
Slovenia	0.01989	−0.04477	0.62234	0.05616	0.85548
Spain	−0.00205	0.06019	0.48567	0.10052	0.34258
Sweden	0.18552	0.08547	1.26139	0.12717	0.37592
Bulgaria	0.12337	−0.00111	1.21367	0.05865	0.39377

Negative variable importance means that the error estimate was higher.

**Table 6 ijerph-19-15091-t006:** Order of Importance of Public Health Expenditures, Private Health Expenditures, and Out-Of-Pocket Health Expenditures for Economic Growth.

lnGOV	lnPRIVATE	lnPOCKET
5.1149	0.9859	2.6149

**Table 7 ijerph-19-15091-t007:** Parameters for SVR Model.

Model	Paremeters	Ranges	Kernel Function	Kernel Scala
SVR	C	[0.1, 5000]	Gaussian	0.27
	ε	[0.0001, 100]		
	γ	[0.001, 50]		

**Table 8 ijerph-19-15091-t008:** Parameters for GPR Model.

Model	Standard Deviation	Kernel Function	Kernel Scala
GPR	0.16	Rational Quadratic	0.27

**Table 9 ijerph-19-15091-t009:** Parameters for DTR Model.

Model	Paremeters	Ranges	Kernel Function	Kernel Scala
DTR	ε	0.021	Quadratic	0.27
	Minimum Leaf Size	12		

**Table 10 ijerph-19-15091-t010:** Results for SVR.

10-Fold	MSE	RMSE	MAE	R^2^ Score
1	0.93881	0.91767	0.90991	0.92355
2	0.87799	0.88198	0.87558	0.89203
3	0.91566	0.87386	0.88366	0.92448
4	0.93141	0.90309	0.89183	0.92739
5	0.91231	0.93684	0.87707	0.87703
6	0.87582	0.89552	0.90236	0.90315
7	0.88677	0.90139	0.88005	0.90527
8	0.87472	0.89143	0.88764	0.88622
9	0.91010	0.87687	0.89677	0.93683
10	0.91613	0.93296	0.92991	0.88778
Average	0.90397	0.90116	0.89317	0.90637

**Table 11 ijerph-19-15091-t011:** Results for GPR.

10-Fold	MSE	RMSE	MAE	R^2^ Score
1	0.81495	0.88828	0.91844	0.84563
2	0.84079	0.84778	0.81955	0.81016
3	0.85704	0.84094	0.82423	0.85394
4	0.84676	0.83874	0.84909	0.90888
5	0.90514	0.85010	0.91085	0.87892
6	0.86738	0.86959	0.81093	0.85012
7	0.86398	0.90643	0.84353	0.90295
8	0.89902	0.84879	0.89832	0.87205
9	0.82691	0.89882	0.82071	0.87787
10	0.88404	0.86075	0.88694	0.88871
Average	0.86060	0.86502	0.85825	0.86892

**Table 12 ijerph-19-15091-t012:** Results for DTR.

10-Fold	MSE	RMSE	MAE	R^2^ Score
1	0.88851	0.83981	0.87651	0.90431
2	0.83928	0.82280	0.85963	0.85359
3	0.87442	0.90679	0.87748	0.81081
4	0.81177	0.82162	0.89078	0.87336
5	0.86913	0.84441	0.89939	0.82560
6	0.82625	0.89900	0.85342	0.83691
7	0.86157	0.89811	0.81648	0.87818
8	0.84879	0.89817	0.88739	0.81523
9	0.87852	0.82113	0.87484	0.91184
10	0.85138	0.85279	0.85729	0.82745
Average	0.85496	0.86046	0.86932	0.85372

## Data Availability

Detailed information about the data set is given in Table 1.

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
