# Peer review of "The Relationship between Health Expenditures and Economic Growth in EU Countries: Empirical Evidence Using Panel Fourier Toda–Yamamoto Causality Test and Regression Models"

_ijerph, 2022, doi:10.3390/ijerph192215091_

Round 1

Reviewer 1 Report

Dear Authors,

The topic discussed in the article is certainly important and up-to-date, but a few issues raised my doubts:

1. no research goal and research hypotheses - there are quantitative research in the article, so the formulation of research hypotheses is definitely justified;

2. the study covered 27 EU countries; these countries are highly diversified and do not form a homogeneous group, it is important not only because of the diversified wealth (which is an important parameter, according to the cited literature on the subject), but also the differentiated health care systems and their financing - in some countries the public care system works efficiently, in others it must be substantially supported by private medical care;

3. referring to point 2 - in some countries, a significant burden of health expenditure is shifted to people who use medical services privately, outside the public health care system. This means that a significant proportion of healthcare expenditure that is incurred in a given country is not included in the study (I understand that only public expenditure is included);

4. there is also no clear correlation between the amount of money spent on health care and its effectiveness, in other words, in other words, higher spending on health care does not mean higher quality and scope of medical services provided, which would translate into the health and productivity of employees, and in as a result - GDP growth. To sum up, an increase in financing does not mean an increase in operating efficiency of the system. This problem is also not raised in the study, but signaled as present in the literature on the subject;

5. it is difficult to understand what justification from the point of view of the study is provided by this fragment: "However some of these countries cannot accept the euro since they ran into issues in the public deficit area during the crisis due to significant government spending. In recent years, these countries were required to reduce their public spending and deficits. " I recommend the authors to analyze the data on the euro area and non-euro area countries in this respect: Public_balance,_2020_and_2021_(¹)_(Net_borrowing_(-)_or_lending_(+)_of_the_general_government_sector,_%_of_GDP)_April2022.png (1000×558) (europa.eu)

Without a wider introduction and justification for this attribute (belonging to the euro area or not), it is difficult to understand what it is all about.

Best regards,

Reviewer

Author Response

Thank you so much for the comments. We appreciate your taking the time.

Reviewer 2 Report

1. English should be improved. There are: grammatical errors; deficient sentences (incomplete or poorly formulated); repetitive expressions (poor vocabulary).

2. It should also be made improvements regarding:

- the presentation of the equations - some are extended over more than one row (e.g. at lines 412-414…)

- the terms are not explained in many of the more complex equations (e.g. lines 298-301...)

3. The content should also be improved in terms of:

- cited references – at least a study cited in the text is missing in the list of references (Yilanci and Gorus, 2020)

- presentation of the methodology – complex equations, but the explanations are briefly given. Perhaps it would be useful to have an Appendix in which the equations are also listed, with explanations, and in the body of the paper more emphasis could be placed on the results obtained and their interpretation.

4. The conclusions are rather brief, and general. The contribution of the authors should be better highlighted here, as well the implications of the results obtained.

Author Response

(The authors gave the same response as above.)

Reviewer 3 Report

The paper approaches a quite known subject of the relationship between public health expenditures and economic growth, which even still debatable is analyzed in the paper using quite simple methods and leading to scarce conclusions. Overall the paper does not emphasize clearly relevant conclusions or the contribution of the authors in this area and they should clarify these aspects and some other issues mentioned below.

It is not entirely clear if the authors use annual data or other kind of data (quarterly, monthly or different). We can suppose that we are talking of annual data, but in this case the sample is quite small for a solid econometric analysis. Moreover, the choice for the variables considered by the authors and the lack of other potential determinants are not clearly explained.

In the part explaining theoretically the methodology, the variables used by PTFY causality test are insufficiently explained in page 7. The same problem appears when explaining the Gaussian Process Regression in page 8.

The choices of the analyzed variables may affect or at least put under question some results such as those of Panel Fourier Causality Test, while authors use GDP per capita at constant prices and current health expenditures, the latter affected by inflation while the first one not. Moreover, beyond observing or not some causalities between the two variables there are no conclusions or comments on the causes of the results obtained or on other determinants on these results, especially while the results are quite different from a country to another and quite equally distributed along the 27 countries.

The results of SVR, GPR and DTR models are insufficiently explained and the models, themselves, which were effectively applied (not the theory on them, but with the specific variables used in the analysis) are not shown. Also, the authors speak of these models as prediction models, but the comments on their results, even very scarce, seem to suggest that they did not predict anything and instead they observed or “demonstrate” some correlations which seems to refer to past and not to future, which is very confusing.

Also, self-citations can and should be avoided.

Author Response

(The authors gave the same response as above.)

Round 2

Reviewer 1 Report

Dear Authors,

Thanks for explanation. Nevertheless, I insist on a precise definition of the goal, not only in the abstract, but also in the article: The aim of this study is to investigate the effect of public health expenditures on economic growth in the period 2000-2019 in 27 European Union (EU) countries.

As you wrote in response to the review: In the study, only the effect of public health expenditures was investigated, the focus of the study is not health expenditures, so the role of the public is a matter of debate, therefore, private health expenditures are not included in the analysis either in the literature.

Best regards,

Reviewer

Author Response

Thank you for the comments. Hope to meet your expectation. Best,

Reviewer 2 Report

1. The paper has undergone some improvements, but there are still some deficiencies in expression (e.g. the expression "according to the study" is repeated too much...)

- "studies are mixed..." (line 93) - maybe the results of studies are mixed

- the sentence in lines 35-36 seems a bit illogical - what's the point? health expenses increase, and birth and mortality rates may also increase... and in the next sentence it is stated that they increase productivity (?)

2. Conclusions should be improved in terms of putting forward the paper results by highlighting their practical, as well as possible topics for future research. At this stage, the conclusions do not make much use of the results obtained - general ideas and too technical results are mentioned - e.g., the value R2 is not of interest, but its interpretation.

Author Response

(The authors gave the same response as above.)

Reviewer 3 Report

Except clarifying the fact that the data are annual, there are very small changes which do not solve the other problems observed in the previous report.

When I raised the issue of the lack of clarification regarding the contribution of the paper I didn't refer at the contribution of each of the many authors to this paper, but to the contribution of the paper to improving the existing literature on the subject, meaning in which way this paper is relevant compared to others and this is one of the still unsolved problems.

Major problems remain regarding explaining the models and, moreover, explaining the results. For instance, there is no mention regarding why to determine MSE, RMSE or MAE and how their values should be interpreted. Also the comments on the results are unclear, they only repeat them without interpreting them and this can not be considered an analysis on data.

Another example is Figure 2, where the authors sustain they present the importance of the independent variables on GDP, but is absolutely unclear how they reached that representation or what is represented on the second scale (the level 3 or 4 or any other means what?, how good or worse is it? 3 is out of 10, or 100 or what?).

Basically, in the results part, out of about 4 pages the comments can be gathered almost entirely on a single page, which is very little for a serious analysis, while considering that many of the comments do not lead to significant conclusions. 

There are also no comments on the limitations of the study, while GDP can be influenced of many other determinants and there remains the problem of comparing constant data with current data.

Author Response

(The authors gave the same response as above.)
